# Anaerobic Performance Profiling in Elite Amateur Boxers

**DOI:** 10.3390/sports12090231

**Published:** 2024-08-27

**Authors:** Tomas Venckunas, Vidas Bruzas, Audrius Snieckus, Loreta Stasiule, Audinga Kniubaite, Mantas Mickevicius, Sigitas Kamandulis, Arvydas Stasiulis

**Affiliations:** Institute of Sport Science and Innovations, Lithuanian Sports University, LT-44221 Kaunas, Lithuania; vidas.bruzas@lsu.lt (V.B.); audrius.snieckus@lsu.lt (A.S.); loreta.stasiule@lsu.lt (L.S.); audinga.kniubaite@lsu.lt (A.K.); mantas.mickevicius@lsu.lt (M.M.); sigitas.kamandulis@lsu.lt (S.K.); arvydas.stasiulis@lsu.lt (A.S.)

**Keywords:** combat sports, amateur boxing, elite athletes, anaerobic capacity, testing, blood lactate

## Abstract

While anaerobic fitness is highly important for the performance in Olympic (amateur) boxing, the relationship between anaerobic performance metrics is poorly understood, and profiling boxers according to their anaerobic capacity is still a challenge. With the current study in elite amateur boxers, we aimed to compare the metabolic and cardiovascular responses to different maximal tests and the intercorrelations between performance indices (peak and mean power, duration of the test, punching frequency) of several all-out tests and their correlation to physiological response metrics (blood lactate and heart rate, HR). Twelve male Olympic boxers performed a battery of tests, including 30 s Wingate cycling and arm cranking, boxing bag punching, steep uphill treadmill running to exhaustion, and progressive treadmill running VO_2_max test. Performance indices of different anaerobic tests were not closely correlated except for the duration of uphill running with body weight scaled (relative) peak and mean power produced during Wingate cycling test and absolute mean power of both Wingate tests. The number of punches landed on a bag per 30 s was associated only with relative power achieved during Wingate cycling test. HRpeak but not peak lactate response correlated strongly across exercise tasks. Finally, no correlation between the highly developed aerobic and anaerobic capacity, suggests a complex picture of the adaptation in elite amateur boxers.

## 1. Introduction

Anaerobic metabolism, the energy turnover without the use of oxygen, is the one of the main sources of energy for active skeletal muscles during high to maximal intensity exercise. It is of crucial importance for performance across various sport disciplines, definitely including Olympic boxing [1,2,3,4], which entails repeated short bursts of high-intensity actions [5,6,7,8]. Such activity makes even maximal rate of oxidative phosphorylation short to supplying all the required energy during exercise leading skeletal muscles to rely greatly on both major anaerobic energy systems, namely ATP-Creatine phosphate and anaerobic glycolysis [9,10].

Performance in anaerobic exercise tests depends, among many other factors such as technical skills and motivation, on muscle size, fiber composition and energy substrate availability [11,12,13], and the power and capacity of anaerobic alactic and lactic energy generating systems being particularly critical [13,14]. Relative contribution by aerobic energy system becomes more significant with extension of the duration of exercise task [14,15,16]. Therefore, even if anaerobic performance is by its own definition independent on the oxygen flux, cardiovascular fitness and aerobic capacity significantly contribute to performance of “anaerobic” exercise tests, and there is evidence it speeds up the recovery from high-intensity exercise bouts [17,18,19,20]. Importantly, all the above factors can be modified by interventions such as exercise training, leading to an improvement in anaerobic power and capacity [21,22,23,24], which is of major relevance to coaches and practitioners aiming to maximize sports performance.

A successful tailoring of training programs requires reliable and valid tools for the assessment of functional state of the athlete. However, accurate assessment of true anaerobic performance potential of boxers remains challenging. Thus, careful selection of suitable tests that would capture metabolic complexities and sport-specific demands is a prerogative. While a 30 s Wingate anaerobic cycling test has been widely used to assess leg peak power and anaerobic capacity in boxers [2,25], the leg work is substantially different from that of during boxing activities. The Wingate test for upper body using arm crank ergometer [26] is a possible alternative or supplement to cycling test, but is evidently poorly representative of the boxer’s punching actions, and it does not induce cardiorespiratory response to the same extent as lower body tests [27].

From the variety of possibilities to assess anaerobic capacity of lower body using all-out tests lasting up to 90 s (e.g., Bosco repeated jumps, upstairs sprinting, repeated running sprints or isokinetic movements in the laboratory), it is Wingate cycling which is used most often for boxers [3,28]. However, it is far from being clear it is the most relevant for boxers, and other tests should be considered as well. An uphill treadmill run test to task failure is particularly appealing in this respect since it is of longer duration than Wingate and most other tests, and somewhat better resembles boxers’ leg movement patterns during contests, in addition to being a body weight bearing activity task. Considering the importance of testing specificity and substantial involvement of upper body in boxing sport, evaluating the utility of a bag-punching test in the context of the above-mentioned lab-based tests is of interest in current boxing research.

The aim of this study in elite amateur boxers was manifold. First, we sought to test for the relationship between performance of different anaerobic capacity tests relevant in the boxing sport context, and these were both upper and lower body protocols including uphill treadmill running, arm cranking, cycling, and punching a boxing bag. Second, we have investigated the correspondence of blood lactate and heart rate (HR) response to these various tests in relation to the maximal HR achieved during the maximal progressive running task. Finally, we looked at the correlation of lower body (running mode) aerobic capacity (VO_2_max) with the performance in various anaerobic capacity tests.

We hypothesized that, in well-trained all-round boxers, there is a significant inter-correlation between anaerobic power, anaerobic capacity and lactate and HR values attained during different anaerobic tests. In addition, we expected that even if metabolic activation (peak blood lactate values) and cardiovascular activation (HRpeak) are lower during the anaerobic tests compared with those attained during treadmill VO_2_max test, there exists a significant correlation between the individual peak values of HR and blood lactate reached during different tasks including anaerobic tests and VO_2_max test. Finally, since the final performance during prolonged sprint or high intensity exercise bouts is partially dependent on the aerobic energy provision, we anticipated that the anaerobic test performance in lower body tests, particularly the uphill running task (due to its longer duration and locomotion specificity), is associated with treadmill running VO_2_max, whereas the performance in upper body anaerobic tests does not show a significant correlation with treadmill VO_2_max.

## 2. Materials and Methods

### 2.1. Organisation and Study Design

In this descriptive, cross-sectional study, we compared physiological markers across four separate all-out exercise tasks in elite amateur boxers: Cunningham–Faulkner anaerobic running test, Wingate arm crank test, and Wingate leg test (performed in the lab), and the maximal bag punching task (done in a boxing gym). The testing sessions were conducted over two consecutive days, with boxers randomly divided into two groups, each completing the two tests per day. The order of these four testing sessions was randomized and counterbalanced among participants. One lower body test (Wingate leg test or Cunningham–Faulkner anaerobic running test) and one upper body test (Wingate arm crank test or bag punching task) were performed on the same day, separated by ~6 h of rest. Additionally, the VO_2_max (treadmill running test) was measured within two weeks of the anaerobic testing sessions.

All the participants read a description of the study before providing their written informed consent for voluntary participation. The study was conducted in alignment with the recent update of the Declaration of Helsinki and was approved by the Lithuanian Sports University Biomedical Research Ethics Committee (No. TRS(M)-29941).

#### 2.1.1. Participants

Twelve male amateur boxers (mean ± SD, range: age, 21.9 ± 4.1, 18–30 years; height, 185.1 ± 7.7, 171–197 cm; body mass, 84.2 ± 14.8, 67.0–109.2 kg), with 9.0 ± 2.6 (range, 4 to 15 years) of training and competitive experience, volunteered for this study. All athletes were from the same club, handled by the national team coach, and all had won medals in the national boxing championships at least once. During the testing period, they were in their active preparatory period, training 1 to 2 times per day for 6 days a week, totaling ~14 h per week over the last 4 weeks. Their training focused on developing power endurance using traditional methods of boxers and without incorporating other forms of high intensity interval training, such as cycling or running exercise modes. For 1 month before and throughout the experiment, none of the participants was on any weight-loss or gain regimens. Boxers were instructed not to perform intense exercise for 2 days before the testing and to keep their diet and daily regimen as stable as possible during the testing period.

#### 2.1.2. Measurements

##### Anaerobic Treadmill Running Test

A short strenuous run on a motorized treadmill (H/P/Cosmos Sports & Medical GMBH, Nußdorf, Germany) was performed following the standardized Cunningham and Faulkner protocol [29,30]. Before the initiation of the test, a warm-up running for 5 min at 10.5 km·h^−1^ with 0% incline was performed and followed by a 3 min rest. To begin the test, the belt speed was set at 12.9 km·h^−1^ and gradient at 20%. Athletes were instructed to run until volitional exhaustion (inability to keep with the treadmill’s belt) and were non-stop vigorously verbally encouraged by the supervising researchers to continue for as long as they can. The time of the test was recorded by a hand-held stopwatch from the moment the athlete fully stepped on the treadmill’s belt by releasing arms off the handrails until he placed his hands back on the support rail after deciding he cannot continue longer.

##### Wingate Cycling Test

The standard 30 s Wingate all-out cycling test was performed on a mechanically-braked ergometer (Monark 824E, Varberg, Sweden) with the brake weight of 7.5% of body weight [31]. Convenient saddle height and handlebar position were individually set, and participants then performed warm-up cycling consisting of 5 min at 60–90 W (1 kg brake weight and pedaling cadence of 60 to 90 rpm) followed by 2–3 (with ~1 min rest) near-maximal effort accelerations of 3–5 s against a brake weight of 7.5% of body weight. Then, after a 2 min rest, the participant started the test by pedaling with no external resistance as furiously as possible after his own decision to initiate the trial. The basket weight was programmed to drop immediately when the pedaling cadence reaches 100 rpm. At that moment the recording of pedaling cadence was started for the exact duration of 30 s. Athletes were given feedback for the elapsed time at 10, 15, 20 and 25 s of the test, and vigorously encouraged by the supervising researchers to pedal as fast as possible throughout the test. The peak power (PP, highest power of 1 s intervals), peak cadence (highest of 1 s intervals), mean power (MP, average over entire 30 s test period), and total energy produced over the 30 s test were recorded, and fatigue index (FI, percentage drop between PP and lowest 1 s interval power) was calculated for the analyses.

##### Wingate Arm Crank Test

To assess the upper body anaerobic capacity, the Wingate 30 s arm crank test was implemented [32] using a brake weight of 4% of body weight. Participant sat on a chair at a comfortable distance in front of the same ergometer (Monark 824E, Varberg, Sweden) securely mounted on a table and equipped with handgrips instead of pedals. The height of the shaft axis was adjusted to be at the chest level. Participants first performed warm-up consisting of 5 min of cranking at 60 W (1 kg brake weight, cadence 60 rpm) followed by 2–3 (with ~1 min rest) near-maximal effort spurts of 3–5 s against a brake weight of 4% of body weight. Then, after ~2 min, the participant on his own decision started the test by cranking the shaft with no external resistance as furiously as possible. The basket weight was again programmed to drop at the cadence of 100 rpm, when the recording started. Athletes were given feedback for the elapsed time at 10, 15, 20 and 25 s of the test, and vigorously encouraged by the supervising researchers to crank as fast as possible throughout the test. The PP, MP, peak cadence, FI and total energy produced were calculated as for Wingate cycling test described above.

#### 2.1.3. Bag Punching Task

Bag punching was conducted in the boxing gym and consisted of one 30 s bout of all-out punching. The task was performed after a standard boxers’ warm-up routine of approximately 20 min, which included low-intensity exercises. Boxers wore their competitive attire and gloves and were instructed to use their usual boxing style while exerting maximally. The task itself was to execute as many forceful straight punches as possible during the first 5 s of the test and throughout the entire 30 s period. The test was video recorded and subsequently analyzed by replaying in slow motion. All punches within 30 s period were counted, and the number of punches thrown during the first 5 s and the last 5 s (Fatigue Index) was calculated.

The custom-made bag used for the punching test was 100 cm in length and 35 cm in diameter dark leather boxing bag stuffed with synthetic textile and weighed 35 kg. It was suspended on a pulley by a rigid steel cable at a 3.5 m height and adjusted so that the middle of the bag was at the chest level of the boxer. To minimize the swinging of the bag during the punching task, the same researcher was holding the bag slightly from the opposite side to keep it in a more stable vertical position.

#### 2.1.4. Incremental Running (VO_2_max) Test

To measure the maximal oxygen uptake (VO_2_max), a ramp treadmill (Lode Katana Sport, Groningen, Netherlands) protocol consisting of a continuous incremental running speed until exhaustion was used. The athletes started the test by jogging at 7·km·h^−1^ for 4 min, and then the speed of the treadmill belt was increased by 0.1 km·h^−1^ every 6 s. Throughout the test, treadmill’s belt gradient was kept constant at 1%, a breath-by-breath gas analysis was performed using MetaMax 3B gas analyzer (Cortex, Leipzig, Germany), and the HR was recorded using an HR meter (Polar H10; Polar, Kempele, Finland). The VO_2_max was considered to be attained when the HR reached >90% of the predicted maximal HR, the respiratory exchange ratio was >1.1, and the participant could not continue running at the required pace, despite encouragement. The VO_2_max and maximal HR during the test were calculated from averaged 20 s intervals. The participants received verbal encouragement throughout the test.

#### 2.1.5. Measurements of HR and Blood Lactate

Heart rate (HR) and blood lactate concentration (La) were measured in response to every implemented test. HR was recorded using a HR monitor (Polar H10; Polar, Kempele, Finland) with HR peak (HRpeak) defined as a highest average over 5 s period of the test. Immediately after each of the five tests the participants assumed supine position on a nearby couch and were resting passively until the blood lactate analysis was over. After carefully cleansing the fingertip skin with an alcohol swab and piercing it with a sterile disposable lancet, 0.3 µL of blood was drawn into a reagent strip of a portable analyzer (Pro2, Arkray Inc., Kyoto, Japan) to measure lactate concentration at baseline (after warm-up), 1 min post exercise and then at 2 min intervals until lactate concentration began declining. The peak lactate response was calculated by subtracting the baseline lactate value from the highest post-exercise lactate level.

### 2.2. Statistical Analysis

Data are presented as means and standard deviation (SD). The Shapiro–Wilk test confirmed that all data were not against the presumption of the normal distribution. Comparison of physiological and performance indices among tests was run using one-way repeated measures ANOVA. In case of statistically significant differences, post-hoc analyses using *t*-tests with Bonferroni correction were performed for pairwise comparisons. Pearson’s correlation coefficient (r) was used to identify relationship between variables, and strength of the relationship was interpreted as follows: moderate, 0.30–0.49; large, 0.50–0.69; very large, 0.70–0.89; nearly perfect, 0.90–0.99; and perfect, 1.00 [33]. The level of significance was set at *p* < 0.05. All statistical analyses were performed using the SPSS Statistics software (v. 22; IBM Corporation, Armonk, NY, USA).

## 3. Results

Descriptive statistics of variables measured during anaerobic and aerobic tests are presented in Table 1. Duration of ART was on average ~50 percent longer than 30 s (the duration of other anaerobic tests) and was in fact > 30 s in each of the participants. Peak power, mean power and total energy were significantly lower in absolute terms (W) for Wingate arm crank test compared to Wingate leg test (*p* < 0.01), while peak cadence attained during the two tests did not differ (*p* > 0.05, Table 1). Similarly, relative values of PP and MP were also significantly lower for Wingate arm crank test compared to Wingate leg test (*p* < 0.01) (Figure 1).

Baseline (pre-test) lactate values were similar (*p* > 0.05) between the tests and averaged 2.0 (0.8), 2.2 (0.7), 2.0 (0.4), 1.9 (0.5) and 1.5 (0.4) mmol/L for ART, WA, WL, PT, and GXT, respectively. Peak blood lactate response after ART was significantly higher than that after all the other tests, all of which had similar lactate increase (*p* > 0.05) (Figure 2).

HRpeak did not differ between ART, WA, WL and PT but all were significantly lower than HR achieved during GXT (Figure 3).

Fatigue index did not differ between WA (55.6 (10.9) %) and WL (53.6 (9.0) %), meanwhile the reduction of the punching frequency was much lower (*p* < 0.01) during bag punching task (19.7 (10.0) %). Peak HR values attained during different tests were significantly strongly intercorrelated (Table 2). Moderate significant correlation between the peak blood lactate response to WL and ART was found (*p* < 0.05). Strong significant correlation between peak blood lactate response to GXT and WA was observed (*p* < 0.05). However, 80 percent of the peak lactate response correlations between the pairs of tests have failed to reach statistical significance (Table 3).

Running duration of ART was nearly perfectly (*p* < 0.01) correlated with the relative PP (r = 0.991, *p* < 0.01), relative MP (r = 0.901, *p* < 0.01) and peak cadence (r = 0.960, *p* < 0.01) and moderately with the fatigue index of Wingate leg test (r = 0.672, *p* < 0.05), while there was no correlation of ART duration with any of the values of Wingate arm test. Absolute mean power during Wingate leg test was significantly correlated with the absolute PP and absolute MP during WA, while the other parameters were not related (Table 4).

Number of punches landed to the bag during the first 5 s of the PT was negatively (r = –0.73 to –0.60, *p* < 0.05) correlated with the absolute PP, relative PP, absolute MP and peak cadence in WA. Total number of punches per 30 s PT was correlated significantly only with relative PP and relative MP achieved during Wingate cycling test (r ≈ 0.70, *p* < 0.05) Fatigue index during PT was not significantly correlated with any variables of ART, WA, WL and GXT.

The absolute VO_2_max was significantly correlated with the absolute PP, relative PP, absolute MP and peak cadence in WA but only with absolute MP in WL (Table 5). The relative VO_2_max was not significantly correlated with any variables of WA, WL, GXT and PT. Absolute VO_2_max (in L/min) was significantly correlated with peak lactate values after the GXT (r = 0.746, *p* = 0.02) but not with HRmax. Running duration of ART was not correlated with VO_2_max.

The interrelationship of the variables of WA are presented in Table 6. The blood lactate concentration was significantly correlated with the absolute PP and MP, relative MP and peak cadence. Peak cadence was significantly correlated with all variables, while HRpeak with any (Table 6).

The interrelationship of the variables of WL are presented in Table 7. Peak blood lactate response correlated with the absolute and relative PP, relative MP, peak cadence and HRpeak. The absolute PP correlated with the absolute MP, peak cadence and FI. The relative PP and peak cadence were significantly correlated with all variables except the absolute MP.

## 4. Discussion

With this study on elite male amateur boxers, we firstly established that performance indices of different anaerobic tests were mainly not well associated except for the correlation of the duration of uphill running test with the relative peak and relative mean power produced during Wingate cycling test and absolute mean power of Wingate arm and Wingate leg tests. Bag punching peak frequency was negatively associated with the power produced during arm cranking (but not leg cycling) Wingate test, indicative of the reliance of heavier blows with more developed upper body musculature. The number of punches per 30 s was associated only with relative power achieved during Wingate cycling test. Secondly, also only partially supporting our initial hypothesis, peak heart rate but not peak lactate response was strongly correlated across different all-out exercise tasks. Finally, and contrary to our initial hypothesis, body weight indexed maximal aerobic power was not correlated to any of the performance parameters of different anaerobic tests.

### 4.1. Anaerobic Performance and VO_2_max

It is tempting to speculate that Wingate anaerobic test performance and VO_2_max is in direct relationship with the level of combat (that is, all-round) athletes [34,35]. The absolute and relative PP and MP values obtained with Wingate cycling test in our study are quite similar to those reported in Polish elite adult male boxers [36], but higher than in Spanish national Olympic team male boxers [37], highly trained Chinese male boxers [38] and elite male table tennis players [39]. Notwithstanding, much lower values for both PP and MP (both relative and absolute) have been reported for elite Turkish male boxers, which could be questioned for reliability since relatively low blood lactate concentrations were reported [40]. We are not aware of studies from other groups investigating the upper body anaerobic capacity of boxers and therefore, cannot much more explicate on this aspect further here. The VO_2_max of our boxers was among the highest reported in combat athletes. Whilst VO_2_max was comparable with values of similarly high caliber senior boxers [4,41], our subjects were heavier on average, thus the high relative aerobic power seems even more impressive. Others reported lower average VO_2_max values either in sub-elite (national level) [2] or elite boxers [42]. Therefore, the above-highlighted findings suggest that, by their comprehensive all-round training, elite amateur boxers are compatible of simultaneously and to a high degree develop both aerobic and anaerobic capacity. In accord, in both power and endurance athletes, changes in anaerobic and aerobic capacity of the lower body are indeed tightly linked [43].

### 4.2. Lactate and Heart Rate (HR) Response

The peak lactate concentration after Wingate cycling test in our subjects was similar to the one reported in Polish and Spanish elite boxers [36,37], but higher than in elite Turkish boxers [40]. In addition, the lactate values we obtained with both Wingate arm cranking and leg cycling tests are higher than those seen in elite male table tennis players [39], but quite similar to those in highly trained combat sport male athletes [28]. Peak lactate average values after the Wingate tests and 30 s duration bag punching (12.5 to 13.5 mmol/L) in the current study were close to those reported after boxing matches in international competitions (13 to 14 mmol/L) [4,44] or simulated match (15 mmol/L) [8]. Average peak lactate after anaerobic uphill running test was the highest (17.5 mmol/L), probably reflecting longer exercise duration of the test and thus time to accumulate end-products of anaerobic glycolysis. Peak blood lactate of average 12.5 mmol/L after VO_2_max treadmill running test in our study and high-level senior Indian boxers [2] was higher than ~10 mmol/L reported by others [36,45]. These underlying differences could be due to not following exactly the same testing protocols, the motivation of the boxers, and macronutrient composition of their diets.

HRpeak achieved during Wingate cycling test in our athletes was lower than that reported in Chinese boxers of similar age [38], but our athletes were of higher competitive level, had more training years behind, were training with larger loads, and had higher aerobic capacity, all of which could lead to a smaller HR response. HRpeak achieved during anaerobic tests in our athletes was lower than that attained during maximal treadmill test to exhaustion, supposed to measure true maximal HR. Therefore, with a single bout of all-out anaerobic exercise task involving either lower or upper body or even both upper and lower body (bag punching task) but lasting only 30 to 60 s, maximal HR (HRmax) is not achieved in highly trained combat athletes. On the other hand, it has been shown recently that HR response is quite close to HRmax during a simulated fight [8]. Summarizing the above, while eliciting pronounced HR increase, anaerobic tests in elite combat sport athletes do not induce activation of cardiovascular system to maximal or even competitive level. Of the anaerobic tests applied, the uphill treadmill run to exhaustion was a sufficient metabolic stimulus of anaerobic glycolysis to increase or surpass blood lactate levels seen in response to boxing matches.

### 4.3. Correlations between Performance and Physiological Variables

As prognosticated, anaerobic capacity measured as a duration of the uphill running test to exhaustion correlated with the relative PP and MP of the cycling Wingate test and fatigue index but not with absolute power developed during cycling. This is because uphill running is a weight bearing activity where relative power generation with the lower limb muscles is of primary importance for performance. Interestingly, uphill running test duration was also quite tightly associated with peak cadence achieved during cycling, which could be indicative of additional importance of “more general” motor function abilities of the legs for the anaerobic function of the lower body muscles during cyclic tasks such as sprint running or cycling.

Arm crank test performance indices were not much associated with the sprint cycling performance markers, except for the absolute cycling MP with arm cranking PP and MP (most likely due to body size effect translation between the test). This suggests the results of the lower body tests should not be viewed as informative of upper body fitness, at least for the all-round trained athletes.

Large to very large correlation between the lactate produced (measured as peak lactate increase from baseline) and most of the power indices of Wingate tests clearly reflects the significance of anaerobic glycolysis to the performance in these tasks. In addition, many of the performance indices were correlated within the test, which is also quite expected since the test is only 30 s in duration and peak power attained is affecting the mean power produced during entire test.

It was interesting to see that while HRpeak attained during the different tests was well correlated, it was largely not the case for the peak lactate response. The explanation could be, the more reactive cardiovascular than metabolic response due to agitation (anticipatory response) associated with the pending efforts [46] and the additional activation of the cardiovascular center within medulla oblongata by the intense firing from motor cortex during exertion when performing the test.

Regarding the specificity of the anaerobic tests used for athletes from different sports, the results of the studies largely show that tests are not interchangeable, and that specificity of the training should be well considered. For instance, in combat sports athletes, specific kicking test has been shown to correlate with Wingate leg test power and lactate values only marginally [47], while neither leg cycling nor arm crank Wingate test performance were related to table tennis-specific anaerobic capacity [39]. Quite interestingly, sports-specific anaerobic work capacity (table tennis activity test) correlated negatively with peak blood lactate concentration after Wingate cycling test but was positively associated with the fatigue index in the Wingate arm crank test, whilst any other Wingate test variables in both ergometer tests did not show significant correlations with sports-specific anaerobic work capacity [39].

Against our hypothesis, aerobic capacity marker relative VO_2_max did not correlate with any anaerobic performance indices, which is indicative of the little overlapping entities measured by these markers of different energetic systems. Indeed high values produced by the anaerobic tests testify for the importance of anaerobic lactic system, which is also relevant to the real-world situations when boxers fight against each other [8,42]. The correlations of VO_2_max in absolute units (L·min^−1^) with absolute power produced during both arm cranking and leg cycling depict the importance of muscle mass involved in the production of these aerobic and anaerobic performance markers, respectively.

In summary, even if uphill running capacity was somewhat linked to power production during Wingate tests and there were some more minor correlations between the power metrics across the tests, performance in different anaerobic tests of elite male amateur boxers is overall weakly correlated. Heart rate but not lactate response correlates well across different all-out exercise tasks, whilst maximal aerobic power was not associated with any of the performance parameters of different anaerobic tests.

## 5. Limitations of the Study

It is of importance that selected resistance (brake weight) for the Wingate anaerobic tests would elicit maximal or nearly maximal power outputs. However, the optimal resistance was neither robustly tailored individually nor previously determined experimentally for this specific group of elite male amateur boxers. Instead, it was arbitrarily selected to be 7.5% and 4.0% of body weight for leg cycling and arm cranking, respectively. While nearly universally using 7.5% break weight for cycling mode Wingate test, researchers are however rather inconsistent in selecting the resistance for arm crank test which varies from 3.5% [48] to 5% [32,49,50] to 8% [32] of body weight. We appreciate the suitability of the applied resistance for a population in our study might be questioned, but some pilot tests revealed that for boxers of this level, and especially so for the athletes of larger weight categories, application of ~4% brake weight gives resistance to convey the cranking quite comfortably, plus the power production is about maximal. Therefore we believe the selected resistance gave a good impression on the upper body anaerobic capacity of these athletes, especially since the relationship between the resistance applied and the powers produced is quite flat [32]. Also, even if the cardiovascular and metabolic response would most likely be different applying higher or lower resistance, we believe the correlation between power outputs and response metrics would remain similar even if somewhat different break weights are used for either leg, arm or both anaerobic tests. Finally, the tests allowed for quite the same average peak cadence, which we see as a validation of properly selected resistance to compare the tests.

Due to limited availability of the athletes, we were restricted to selecting few of the tests without probing for more options from the arsenal of existing anaerobic tests. However, the tests we selected covered for the muscles of both upper and lower body as well as a specific task of bag punching where upper body is presumably mostly involved while leg muscles also significantly contribute to the power produced. We have to admit it as a clear shortcoming of the study that the bag-punching task, because of the limitation of the equipment, allowed us to analyze only the punching frequency but not impact forces.

Some other limitations include the lack of muscle anaerobic enzyme activity measurements, which would however require invasion into the tissues, thus limiting us to indirect indices of estimation of anaerobic lactic metabolism, where we selected one of the most common but still a single marker. In addition, we have limited our analyses to senior male amateur boxers, which does not allow extrapolation of the findings to other combat sport populations, not to mention the athletes from other sports or untrained individuals. Finally, a small number of subjects plus their spread between weight categories might have influenced the working capacity presented as well as physiological responses and correlations detected.

## 6. Conclusions

Performance and physiological strain markers correlated more consistently and stronger between the anaerobic tests of the same muscle groups than between tests for different body regions. Therefore, anaerobic tests for different muscle groups should not be used interchangeably but rather carefully selected to represent the aimed characteristic of the athlete. No correlation between anaerobic performance with aerobic capacity, even if both are well developed in this particular population of all-round sport elite athletes, suggests a complex picture of the adaptations occurring. Thus, identifying strengths and weaknesses of aerobic and anaerobic energetic systems, then adjusting their training plans and monitoring them by regular testing during their training could be one of the approaches for the tailoring the individualized development and tuning of energetic systems in boxers.

## Figures and Tables

**Figure 1 sports-12-00231-f001:**
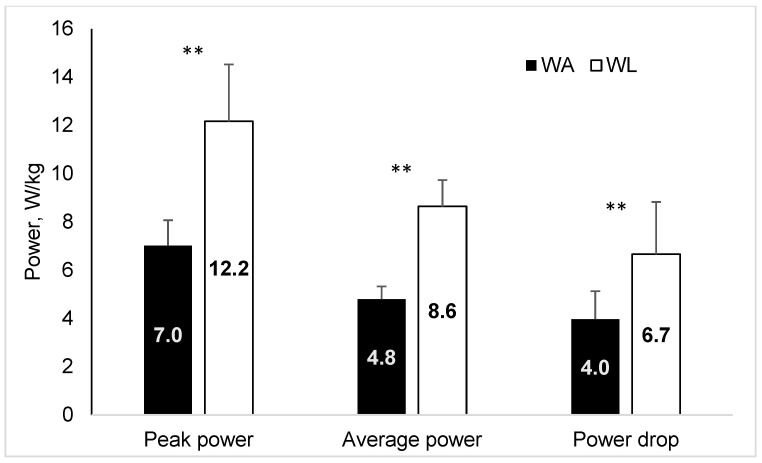
Peak and mean power relative to body weight produced during Wingate cycling (WL) and arm cranking (WA) (** *p* < 0.01 between the WL and WA tests).

**Figure 2 sports-12-00231-f002:**
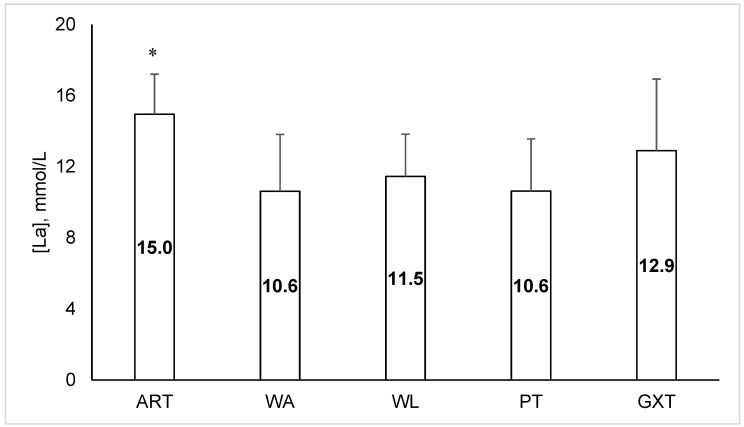
Peak blood lactate ([La]) increased from baseline in response to different exercise tests (ART, Cunningham–Faulkner anaerobic running test; WA, Wingate arm crank test; WL, Wingate cycling test; PT, bag punching task; GXT, graded running test; * *p* < 0.05 compared to all other tests).

**Figure 3 sports-12-00231-f003:**
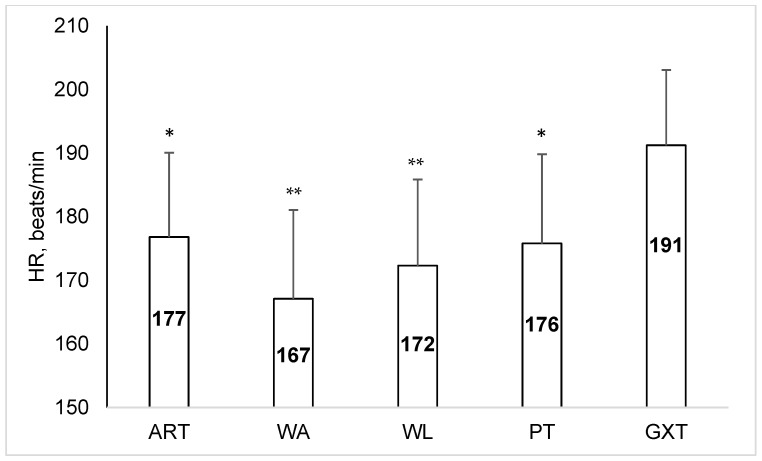
Peak heart rate (HRpeak) values achieved during different exercise tests (ART, Cunningham–Faulkner anaerobic running test; WA, Wingate arm crank test; WL, Wingate cycling test; PT, bag punching task; GXT, graded running test; *, ** *p* < 0.05 and *p* < 0.01 compared to GXT).

**Table 1 sports-12-00231-t001:** Descriptive statistics of variables measured during anaerobic and aerobic tests (ART, Cunningham–Faulkner anaerobic running test; WA, Wingate arm crank test; WL, Wingate cycling test; PT, bag punching task; GXT, graded running test; ** lower than in WL at *p* < 0.01).

		Mean	Standard Deviation	Range
ART	Running duration, s	45.3	10.6	34–65
WA	Peak power, W	593.4 **	156.7	430.2–890.2
	Mean power, W	404.7 **	90.7	285.9–570.8
	Peak cadence, rpm	144.7	15.2	120.0–169.0
	Total energy, J	11,911 **	2617	8437–16,867
WL	Peak power, W	1008.0	168.6	721.2–1231.4
	Mean power, W	719.9	107.5	595.0–889.5
	Peak cadence, rpm	145.2	17.6	122.9–170.6
	Total energy, J	21,064	3057	17,281–25,852
PT	Number of punches during 1–5 s	28.6	2.5	25.0–31.0
	Number of punches during 26–30 s	22.9	2.6	17.0–26.0
	Number of punches per 30 s	162.7	23.0	124–191
GXT	VO_2_max, L·min^−1^	5.23	0.83	4.03–6.53
	VO_2_max, mL·kg^−1^·min^−1^	61.5	5.3	53.8–72.1

**Table 2 sports-12-00231-t002:** Correlations between peak heart rate (HR) of different exercise tests (ART—Cunningham–Faulkner anaerobic running test; WA—Wingate arm crank test; WL—Wingate cycling test; PT—bag punching task; GXT—graded running test; * *p* < 0.05; ** *p* < 0.01).

	ART	WA	WL	GXT
WA	**0.874 ****			
WL	**0.936 ****	**0.942 ****		
GXT	**0.836 ****	**0.794 ***	**0.796 ***	
PT	**0.835 ****	**0.891 ****	**0.876 ****	**0.850 ****

Note. Significant correlations are highlighted in bold.

**Table 3 sports-12-00231-t003:** Correlation between peak blood lactate response to different exercise tests (ART—Cunningham–Faulkner anaerobic running test; WA—Wingate arm crank test; WL—Wingate cycling test; PT—bag punching task; GXT—graded running test; * *p* < 0.05).

	ART	WA	WL	GXT
WA	0.360			
WL	**0.631 ***	0.277		
GXT	–0.155	**0.730 ***	–0.095	
PT	0.392	0.520	0.450	0.173

Note. Significant correlations are highlighted in bold.

**Table 4 sports-12-00231-t004:** Correlations between variables of Wingate arm crank test and Wingate cycling test (* *p* < 0.05; ** *p* < 0.01).

		Wingate Arm Crank Test					
		Absolute PP	Relative PP	Absolute MP	Relative MP	Peak Cadence	Fatigue Index
**Wingate Cycling test**	Absolute PP	0.455	0.317	0.513	0.371	0.355	0.130
	Relative PP	–0.303	0.100	–0.330	0.168	0.023	–0.002
	Absolute MP	**0.613 ***	0.277	**0.782 ****	0.477	0.395	0.033
	Relative MP	–0.326	0.071	−0.248	0.335	0.047	–0.133
	Peak cadence	–0.205	0.165	–0.232	0.217	0.112	0.101
	Fatigue index	–0.139	–0.129	–0.170	–0.176	–0.168	–0.114

Notes. PP, peak power; MP, mean power. Significant correlations are highlighted in bold.

**Table 5 sports-12-00231-t005:** Correlations of VO_2_max with variables of Cunningham–Faulkner anaerobic running test (ART), Wingate arm crank test (WA) and Wingate cycling test (WL) (* *p* < 0.05; ** *p* < 0.01).

		Absolute VO_2_max	Relative VO_2_max
**ART**	Running duration	–0.457	0.467
**WA**	Absolute PP	**0.906 ****	–0.234
	Relative PP	**0.726 ***	0.287
	Absolute MP	**0.828 ****	–0.279
	Relative MP	0.432	0.301
	Peak cadence	**0.768 ****	0.203
	Fatigue index	0.366	–0.304
**WL**	Absolute PP	0.441	0.061
	Relative PP	–0.366	0.465
	Absolute MP	**0.639 ***	–0.174
	Relative MP	–0.364	0.442
	Peak cadence	–0.269	0.409
	Fatigue index	–0.037	0.320

Notes. PP, peak power; MP, mean power. Significant correlations are highlighted in bold.

**Table 6 sports-12-00231-t006:** Correlations between variables of Wingate arm crank test.

	[La]	Absolute PP	Relative PP	Absolute MP	Relative MP	Peak Cadence	Fatigue Index
Absolute PP	**0.658 ***						
Relative PP	0.560	**0.752 ***					
Absolute MP	**0.785 ****	**0.904 ****	0.537				
Relative MP	**0.755 ***	0.541	**0.741 ***	0.623			
Peak cadence	**0.650 ***	**0.800 ****	**0.974 ****	**0.648 ***	**0.802 ****		
Fatigue index	0.096	0.479	0.607	0.151	0.043	0.560	
HRpeak	0.047	0.195	0.407	0.188	0.532	0.430	0.078

Notes. [La], peak blood lactate response; PP, peak power; MP, mean power; HRpeak, peak heart rate. * *p* < 0.05; ** *p* < 0.01. Significant correlations are highlighted in bold.

**Table 7 sports-12-00231-t007:** Correlations between variables of Wingate cycling test.

	[La]	Absolute PP	Relative PP	Absolute MP	Relative MP	Peak Cadence	Fatigue Index
Absolute PP	**0.693 ***						
Relative PP	**0.805 ****	0.548					
Absolute MP	0.513	**0.851 ****	0.130				
Relative MP	**0.889 ****	0.483	**0.903 ****	0.247			
Peak cadence	**0.859 ****	**0.633 ***	**0.984 ****	0.248	**0.909 ****		
Fatigue index	0.439	**0.675 ***	**0.733 ***	0.316	0.557	**0.725 ***	
HRpeak	**0.753 ***	0.506	**0.684 ***	0.363	**0.791 ****	**0.736 ***	0.507

Note. [La], peak blood lactate response; PP, peak power; MP, mean power; HRpeak, peak heart rate. * *p* < 0.05; ** *p* < 0.01. Significant correlations highlighted in bold.

## Data Availability

The raw data presented in this study are available on reasonable request from the corresponding author.

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
