# Peer review of "Anaerobic Performance Profiling in Elite Amateur Boxers"

_sports, 2024, doi:10.3390/sports12090231_

Round 1
Reviewer 1 Report
Comments and Suggestions for Authors
You should indicate what type of boxing, since Olympic boxing is not the same as professional boxing and there are several international federations, in adition to the fact that the physiological requirements are unequal.
It is indicated that they have trained in power resistance work, but not whether or not they are used to high intensity intervallic training specifically, this would help to correctly assess the starting point of their fitness level.
It would be interesting if they had devised a new protocol; the Wingate test does not technically resemble the punching action. It should be indicated what protocol (if any) they have proposed for recovery between sessions of days or hours.
The tests have a consistent bias in that only in the bag test the movement is functional and integrated, the rest of the tests only contemplate the lower limb, there should be some review in this regard.
There are differences in intensity between 90% of heart rate and 95 or 98%, the target value is VO2max. They have not considered taking the reserve heart rate? in fact, it does not correlate with the rest of the variables measured.
In the discussion, it is missing to establish relationships with other adversary sports as they have done with table tennis, in order to give it greater applicability (fencing, taekwondo, etc.). In the argument, they suggest that the form in which the upper limb is found is the most important. This is debatable, because although the striking is done with the hands, the kinetic chain is global. If I have misinterpreted it, it should be rewritten to avoid confusion.
Author Response
We have uploaded responses in the attached file.

Reviewer 2 Report
Comments and Suggestions for Authors
Please find attached

Comments on the Quality of English LanguageMinor issues with the English language that can be easily addressed.
Author Response

(The authors gave the same response as above.)
